# Multi-Stakeholder Retrospective Acceptability of a Peer Support Intervention for Exercise Referral

**DOI:** 10.3390/ijerph18041720

**Published:** 2021-02-10

**Authors:** Robert M. Portman, Andrew R. Levy, Anthony J. Maher, Stuart J. Fairclough

**Affiliations:** 1Health Research Institute, Edge Hill University, Ormskirk L39 4QP, UK; levya@edgehill.ac.uk (A.R.L.); faircls@edgehill.ac.uk (S.J.F.); 2Department of Sport and Physical Activity, Edge Hill University, Ormskirk L39 4QP, UK; mahera@edgehill.ac.uk; 3Department of Psychology, Edge Hill University, Ormskirk L39 4QP, UK; 4Institute for Social Responsibility, Politics, Pedagogy and Practice in PE and Sport Research Group, Edge Hill University, Ormskirk L39 4QP, UK

**Keywords:** peer support, social support, exercise referral, qualitative

## Abstract

Perceived social support opportunities are central to successful exercise referral scheme (ERS) client experiences. However, there remains a lack of guidance on how ERSs can embed social support opportunities within their provision. This study presents retrospective acceptability findings from a 12-week social-identity-informed peer support intervention to enhance perceived social support among clients of an English ERS. Five peer volunteers were recruited, trained, and deployed in supervised ERS sessions across two sites. Peers assisted exercise referral officers (EROs) by providing supplementary practical, informational, motivational, and emotional support to ERS clients. Individual semi-structured interviews were conducted with peers (*n* = 4), EROs (*n* = 2), and clients (*n* = 5) and analysed thematically. The analysis identified three primary themes. The first theme detailed how EROs utilised peer volunteers to supplement the ERS client experience. This theme delineated peer roles within the ERS context and identified salient individual peer characteristics that contributed to their success. The second theme described peer acceptability among the various stakeholders. Peers were valued for their ability to reduce burden on EROs and to enhance perceptions of comfort among ERS clients. The final theme presented participant feedback regarding how the intervention may be further refined and enhanced. Peers represented a cost-effective and acceptable means of providing auxiliary social support to ERS clients. Moving forward, the structured integration of peers can improve the accessibility of social support among ERS participants, thus facilitating better rates of ERS completion.

## 1. Introduction

The effectiveness and cost-effectiveness of exercise referral schemes (ERSs) are considered to be poor [1]. Scheme dropout rates are high [1], and completion is associated with only modest and variable improvements in physical activity (PA) and health-related outcomes [2,3]. The evidence base for ERSs is currently limited by improper recognition of schemes as representing a singular model of intervention. In practice, ERSs are vastly heterogenous in respect of scheme length, setting, and delivery format, as well as the individual sociodemographic characteristics of participants who access them [3,4]. In response, Hanson et al. [5] proposed a taxonomy to improve the accuracy of ERS classification, thus enabling future investigation of variable ERS effectiveness according to salient delivery criteria. Moving forward, the use of the taxonomy may facilitate widespread promotion and adoption of effective delivery strategies across the ERS landscape. Notwithstanding, there already exists a plethora of qualitative research to document the primary facilitators and barriers to ERS completion [6,7,8], without comparable evidence to indicate improved rates of ERS completion [9]. Taken collectively, these findings suggest inability to successfully translate increased awareness of facilitators and barriers to ERS practice. In this regard, social support is consistently associated with better ERS engagement [10,11,12]. ERS participants frequently emphasise the importance of receiving social support [11,13,14], and express dissatisfaction when social support is perceived to be lacking [12,15,16]. Nonetheless, there remains uncertainty regarding the extent to which social support opportunities are explicitly embedded within ERS provision, with perceived accessibility of social support varying according to demographic [11] and personal [12] participant characteristics, such as age, gender, and social anxiety. Thus, there remains a need to identify effective strategies to provide social support among ERS participants who are eager but currently unable to access it. 

Haslam et al. [17] championed the subjective experience of social identification as the primary mechanism for facilitating reciprocal social influence, underpinning both the provision and receipt of social support. Where individuals perceive themselves to share category group membership with similar others, they are more likely to experience a sense of belonging and social connectivity [18,19]. Moreover, shared social identity positively predicts the perception of social support over and above the frequency or volume of available support [17]. Accordingly, though ERSs are traditionally delivered within densely populated exercise environments, opportunities to develop and adopt shared social identity may be insufficient. The structured integration of a peer-based social support intervention presents an opportunity to enhance perceptions of belonging and social identification among ERS participants, thus enhancing perceptions of social support. Subsequently, improved understanding of optimal social support provision can increase the likelihood of ERS completion and improve the effectiveness and cost-effectiveness of ERSs. Peer-based interventions can also yield positive health-related outcomes, such as increased PA [20] and well-being [21]. Peer-based interventions are flexible, facilitating the development of bespoke peer support strategies to address specific contextual needs. Identification, development, and promotion of contextually appropriate peer roles and establishing quality relationships between peers and intended recipients of peer support are distinctive hallmarks of effective peer-based interventions [22,23]. Perceptions of similarity between peers and recipients of peer support are critical to peer acceptability and commonly achieved via the use of age- and/or gender-matching peer assignment strategies [24,25]. However, there remains a lack of evidence to document the development of a structured ERS peer support intervention. Subsequently, there is a current lack of understanding of salient factors contributing to the development of positive peer support relationships within an ERS context, and a corresponding dearth of knowledge regarding the acceptability of a peer support intervention among various ERS stakeholders. This study presents retrospective acceptability findings of a 12-week peer support ERS intervention, exploring salient factors linked to the successful implementation of peer support within this context. Specifically, this study sought to identify (1) how ERS staff and clients utilised peer support volunteers; (2) the fidelity of the peers performing roles as intended; (3) retrospective acceptability among clients, exercise referral officers (EROs), and peers; and (4) practical considerations to inform future peer implementation across ERSs.

## 2. Materials and Methods 

### 2.1. The ERS Context

The ERS consisted of a 12-week gym-based intervention delivered free of charge across four community gyms in the West Lancashire region of northwest England. Clients gained access to the scheme via referral from a primary or secondary care organization (e.g., by a general practitioner (GP)) or “self-referral.” At each site-specific delivery location, clients were given a timetable providing details of drop-in sessions available at different times throughout the week (e.g., Monday and Thursday, 2:00 p.m.–4:00 p.m.; Wednesday, 11:30 a.m.–1:30 p.m.). Drop-in sessions were supervised by EROs qualified to minimum level 3 Register of Exercise Professionals. Supervised sessions comprised neither group-based nor strictly one-to-one-based provision. Instead, EROs provided support to multiple ERS clients intermittently throughout the duration of supervised sessions. Supervised sessions were also accessible to non-ERS community gym users. EROs provided all ERS clients with training programmes bespoke to their health-related needs and current fitness levels. Training programmes exclusively consisted of the use of cardiovascular endurance or resistance-based machines typically found within gyms, such as treadmills, cross-trainers, and fixed weight machines.

### 2.2. The Peer Support Intervention

EROs and previous clients took part in a qualitative prospective acceptability study prior to the development of the peer intervention [26]. The prospective acceptability study was underpinned by a social identity approach and sought to understand salient demographic and personal characteristics of ERS peers and desirable peer roles. Findings from the prospective acceptability study directly informed recruitment and training procedures for the peer intervention. Subsequently, four individuals who had previously completed the ERS were recruited, trained, and positioned as peer volunteers within supervised sessions across two leisure sites offering ERS in September 2019. One additional peer was recruited in November 2019 to cover an extended peer absence. Peers were integrated within supervised ERS sessions under the guidance of designated EROs (*n* = 2). Each peer agreed to assist EROs for two 1-hour sessions per week for 12 weeks. The peers were specifically instructed to provide the following forms of support to ERS clients: (1) practical (e.g., nonspecialist assistance using exercise equipment, such as helping to change the resistance settings on weight machines or to show clients how to access preset programmes on cardiovascular endurance machines), (2) informational (e.g., general guidance on exercise and gym facilities), (3) motivational (e.g., positive messaging), and (4) emotional (e.g., helping clients to feel at ease within the exercise environment) [26]. The peers were explicitly informed not to attempt to provide any specialist advice on how to use or operate gym equipment, such as providing recommendations on exercise type, technique, or intensity. The peers wore a branded “peer volunteer” T-shirt during peer support sessions. The peers were incentivised via free access to the leisure site facilities for the duration of their time as peers and a free 6-month membership following the end of their involvement. 

### 2.3. Participants

Participant eligibility comprised peers (*n* = 4), EROs involved in the recruitment and management of the peers (*n* = 2), and current or recent ERS clients who received peer support (*n* = 5). Peer and client ages ranged from 44 to 67 years (58.8 ± 8.4 years) and 49 to 77 years (64.4 ± 10.76 years), respectively. The key demographic characteristics of the peers and clients are presented below (Table 1). EROs’ ages ranged from 42 to 49 years (45.5 ± 3.5 years), and all the participants identified as White British. 

### 2.4. Data Collection

Institutional ethical approval was sought and obtained prior to data collection (SPA- REC-2017-008), which commenced between November 2019 and March 2020. The peers and clients provided written consent allowing the use of descriptive demographic characteristic data upon ERS sign-up as part of an ongoing quantitative evaluation. All the participants provided additional written and verbal consent specific to the interview process prior to the interview. The EROs were asked to identify and recruit clients who received peer support at any stage of their ERS involvement before contacting the researcher to arrange an interview. The interview length for the peers and clients ranged from 21 to 42 min (mean duration = 30 min) and 28 to 46 min (mean duration = 35 min), respectively. ERO interviews had 37- and 33-min duration, respectively. The peers and EROs were interviewed 6 weeks following the initial introduction of ERS peer sessions. Clients were interviewed at various stages of their ERS participation ranging from 4 to 10 weeks. Interview guides were developed in the pursuit of attaining experiential understanding of the roles of peers within the ERS context from the perspectives of clients, EROs, and peers. Interview guides enabled the capture of the manner in which the clients were first exposed to peer support, if/how interactions with peers changed over time, and the impact of peer support on social and general ERS behaviour. The interview guides encompassed peer roles and responsibilities, the implementation of peers within supervised sessions, and the retrospective acceptability of peers. The wording of the interview guides varied according to participant status as a client, ERO, or peer, though the focus of the questioning remained consistent. For instance, the peers and EROs were asked questions pertaining to the process of welcoming new clients to the scheme, whereas the clients were asked to detail their initial experiences interacting with peers. Triangulation of participant responses during data analysis facilitated intricate understanding of peer roles.

### 2.5. Data Analysis 

Data were analysed thematically using the guidance set out by Braun and Clarke [27]. A principally deductive analytic approach was employed to address the stated study aims. Specifically, data were analysed in a manner that identified (1) the roles performed by peers within the ERS context; (2) the extent to which clients, EROs, and peers perceived these roles to be valuable; and (3) the ways in which the ERS peer intervention may be refined to promote future success. First, data immersion was achieved by conducting interviews, listening to audio files and transcription of audio recordings, and reading and rereading transcripts. Engagement in these processes facilitated an intimate familiarity with data, contributing to subsequent sense-making and knowledge construction. Second, chunks of the data set were assigned codes in relation to specific study aims. At this stage, chunks of the data set that contained interesting and/or unanticipated perspectives were also coded. The third and fourth steps involved the identification of codes that reflected similar issues before pooling these codes together to construct potential themes. These initial themes were scrutinized for suitability and consistency to ensure each code was appropriately reflected by its overarching theme. Once the final thematic structure was established, illustrative quotes were extracted from the data set as supporting evidence.

## 3. Results

Three main themes are presented to convey retrospective acceptability findings from the peer support ERS intervention. The first theme delineates how EROs utilized peers to support ERS provision. Additionally, subthemes are presented to detail the roles peers performed and individual differences in peer approaches. The second theme documents perceived benefits of peer support from the perspectives of EROs, clients, and peers. The participants discussed overall peer acceptability and how peer roles and demographic and personal peer characteristics influenced perceptions of acceptability. The third theme centres on participant feedback on how the peer intervention could be enhanced. The participants provided feedback related to the timing of the intervention, factors influencing the development of positive peer-recipient relationships, and proposed modifications to the peer training programme. 

### 3.1. Utilising Peers within ERS

This theme presents an overview of how peers were utilised by EROs within the ERS context, details common peer roles, and provides examples of individual peer approaches. Sigourney (ERO1) discussed how she introduced peers to new clients and socially integrated them within the ERS environment:
*I introduce the volunteers, explain what the volunteer role is to that person* [client]. *Get them to know each other. So, I would stay with the volunteer and that participant until they’d got a bit of a rapport going, and then I’d go off and do things with other people that were in the gym.*

Georgia (ERO2) adopted a similar practice:*“I would introduce the volunteer* [to a new client] *immediately, possibly before I’ve even left this room* [where initial assessment appointments took place], *because the volunteer would already be around. And then just integrate them into chatting.”* Accordingly, EROs utilised peers as supplementary early sources of general and social support for new clients. Mabel (P1) corroborated ERO accounts: “[The ERO], *I’d say, has introduced everyone to me, and introduced me to them. When they first arrive, she’ll always say, ‘Oh, have you met* [Mabel]*? She’s one of our volunteers.’*” As did Jessica (C4):
I think she [ERO] just introduced me and said if she wasn’t there or she was with somebody else and they were there, to ask any questions, or help if she wasn’t available or something like that perhaps. And then, you know, they started talking to me and stuff and, that was it.

Sigourney (ERO1) discussed designating peer support based on the severity of a client’s medical condition: “*I will sort of… if I know say that* [client] *is coming in, I will sort of allocate* [peer] *to him a little bit more. So, I can concentrate maybe on someone that’s higher needs than what* [client] *is.*” Georgia (ERO2) recounted an experience where she had used peers in a similar capacity: “*I had a time where I had a very high-needs client, and I had to have them in here just for a little moment, and again*, [the peer] *just took over with the people that were out there.*”

#### 3.1.1. Peer Roles

The clients were asked to share their experiences of peer support during supervised sessions. As explained by Agnes (C5), peers commonly assisted clients in their use of exercise machines/apparatus:
It’s probably age-related, but these pin things, you know (laughs), I often think to myself, “Now, do I pull these ones in and keep these ones out? Or do I pull these ones out and keep these ones in?” So, things like that. So, erm… they’re on hand then… you know, to… for me to just be able to shout over and say… “Can you just remind me which pins I need to pull” (laughs).

Ricky (P3) described a related example where he had offered non-technical guidance to a client relating to the use of the exercise machines:
*I have this boiled kettle scenario. Across the top of the television screens, it has your heartbeat… calories, heartbeat, time, blah-di-blah. Now I found over a period of time that I didn’t like that, and then I turned around one day to [ERO], and I was just saying to him about it. He said, “Well, yeah, turn it off.” So, I said to some of them* [clients] *now, “Turn it off.” It’s not like watching a kettle boil then. You can just sit and watch television. Try and get it into your [head] what you’re doing, and just keep pressing it every now and then.*

Unfortunately, there were isolated examples where peers appeared to recommend or encourage clients to try new exercise machines. Agnes (C5) positively described how a peer had been helpful in showing her how to navigate an unfamiliar piece of equipment:
*The second time I was here, erm… there was a… erm… [an exercise machine] at the end, the one where you have to put your hands back like this, and your elbows up like this. Well, I had not been on that, and I thought, “Oh, that looks quite good that.” You know. Erm… so, when* [the peer] *was coming, and said to me, “Oh, have you been on that?” and I said, “No, I haven’t,” so they came across and like, I went on that, and they made sure that I was doing it correct.*

Sigourney (ERO1) acknowledged she had found peers to be engaging in a similar behaviour: “*At first,* [the peer] *was tending to try and teach people to use certain machines that I had not put them on.*” However, it was explicitly emphasised in the training package that peers should not attempt to offer any form of prescriptive or technical advice related to the selection, technique, or intensity of clients’ exercise behaviour. Positively, Sigourney (ERO1) continued to describe that this behaviour had been successfully quashed: “*We’ve addressed that all now. That’s all sorted. So, they do know not to do that, or up weights for people.*”

Clients also discussed ways in which peers offered social support during their time on the scheme. Jessica (C4) talked about how peers immediately welcomed her upon arrival to the gym, before circling back around at a later point for a chat:
As soon as I came in, he just said, “Oh, hi, are you OK?” and stuff, and then I just went on the machine. And then normally, he will come and talk to me later on. He usually comes to chat to me when he’s free.

Agnes (C5) recounted a similar experience:
*As soon as I came in today and went on the treadmill, I think… I don’t know, is it* [Mabel]*?… I get mixed up with the names because there’s so many of them, the lady volunteer we’ll say, erm… came right across and said, you know, “how was I doing.” You know, so that’s nice. That’s nice, yeah. Oh, they do interact very well, the volunteers. You know, they go around to the different people and just have a chat and make sure that everything’s going alright.*

The peers and clients were asked to divulge common topics to arise during their social exchanges. Conversations between peers and clients encompassed a wide range of topics, covering politics, sport, and the weather. The peers appeared to especially value conversation topics that were unrelated to exercise or health and well-being:
There’s been a lot of topical stuff. Like, the Friday was taken up with the… the election was the day before, so there was a lot of talk about that. Yeah, people tell you about their interests, tell you about their plans. Yeah. No, it’s a really good mix of… it’s not all about, it’s definitely not all about the gym and about the exercise.(Carl: P4)
*When they’re not sitting talking about gym, gym, gym, I think I’m doing a decent job. Like… one lady the other day who was in, she said, “What did you think of the final? Did you think it was all right?” And that was the first thing she said to me. Didn’t even say, “Hello,* [Ricky]*,” or anything. “Did you watch the final?” and that was “I’m a Celebrity, Get Me Out of Here.”*(Ricky: P3)

However, Ricky (P3) went on to explain how these social exchanges could then segue into a discussion of the client’s health and well-being: “*And she’s there pedalling away, you know. ‘Did you watch the final? Oh, yeah,* [what] *did you think…? Oh, yeah.’ And, you now sit down and, ‘How are you doing? How’s it feeling? How’s… Is it good?’*” 

#### 3.1.2. Individual Peer Approaches

Within discussions relating to the content of social support, it became apparent how the diverging personal and professional backgrounds of the peers shaped their approach to the role. Sigourney (ERO1) discussed the different approaches adopted by the peers under her management:
[Carl] *is really good on the mental health side of it. Because he knows… because he’s going through that himself. He knows how to sort of approach it, probably in a better way than me, to be honest with you. He’s probably really good at that.* [Mabel]*’s more the have a giggly type one, so if you’ve got somebody who’s a bit quieter, pulling them out of that background.* [Paul]*’s more reserved, but he is good at being able to sort of encourage people that are lacking in confidence. So, he’d say, “Oh, when I started, I was x, y, and z, and now I’m like, you know I’ve lost weight, I’ve done this, I’ve done that.” So, I think* [Paul]*is a good sort of motivator in that sense of “Look what I’ve achieved.”*

Overall, these differences were reflected in the approaches of the peers themselves. The peers were divided in their apparent comfort when engaging new clients. Mabel (P1) described her initial conservative approach: “*I’m a bit of a people watcher. Just watch. ‘Do they need help? Does it look like they want to have a chat?’ I don’t go in full on and say, ‘Who are you?’ and you know, anything like that.*” In contrast, Ricky (P3) was much more forthcoming: “*I come in here with the lad, swimming* [when not acting as a peer]*, and as people’ll walk out, ‘Oh,* [Ricky],’ *and they know me, and it’s through me pushing my face here, there, and everywhere. I’m great at pushing my face in things.*” Ricky’s (P3) professional background as a taxi driver belied his ability to socially engage clients: “*My way of talking to people when they’re in a taxi doesn’t differ a great deal to the way I talk to people in the gym, but I wouldn’t start talking about my medical conditions to people in my taxi!*” Alternatively, Paul (P2), a retired psychiatric nurse, described how his prior professional experiences imbued his own approach to the role, also believing clients to be more receptive to his support after sharing details of his professional background with clients:
Once they get to know me and they seem more relaxed, they tend to talk about why they’re here. What… people talk about their health conditions as well. Because I let them know that I’m an ex-nurse. And I think that helps as well. Because they can tell me things that they might not be able to tell other people.

The provided peer recruitment guidelines focussed on identifying individuals who fitted a particular demographic and personality profile. However, peers were sometimes able to utilise suitable aspects of their own personal and professional experiences to successfully engage clients in different ways.

### 3.2. The Benefits of Peer Support

Georgia (ERO2) praised the supportive nature of the peers: “*They’ve*[peers]*been very enthusiastic and very supportive and able to… kind of reach out to our new customers, which has been quite nice*.” Sigourney (ERO1) considered clients to be receptive to peers, recounting examples where clients would ask about the whereabouts of peers if they were not in sessions: “*When they’ve come in and they’ve missed* [Carl] *by 5 min or whatever, it’s they’ve actually been asking like, ‘Where is he?*’” Clients were similarly receptive to the support offered by peers. Jessica (C4) believed the peers had helped her feel more comfortable:
It’s nice that they are so friendly and welcoming because it is a bit intimidating sometimes, isn’t it, going to a gym? So, it is nice when you walk in and someone says… and they know your name as well, and they just go… you know, you kind of feel like you should be there.

Peers themselves perceived their role to be beneficial to clients, often recounting anecdotal or observational evidence to support this view. As expressed by Paul (P2), “*They’ve* [clients] *said that they’ve enjoyed the sessions that they’ve attended. It’s been beneficial to them to have somebody to show them what to do.*” Similarly, Mabel (P1) recounted an experience where a client shared “surprising” personal information with her:
I just went over to him and said, “Oh, I believe you’re a bit of a golfer.” You know, and I do know a bit because my family play it. And then he started telling me about his op. And then I was speaking to his wife later on. And I said, “Oh, he said he hadn’t been so good,” and she said, “Oh, I’m so shocked that he’s told you that. It’s supposed to be in the family. He’s not supposed to tell anybody that” (laughs). But she meant it in a nice way, that he’d actually spoke about it.

These experiences were central to peers’ own sense of fulfilment for the role. As articulated by Ricky (P3), “*I enjoy doing it. I get pleasure out of it, and if I’m passing a little bit on to somebody else… even if it’s only a little bit… it’s better than nothing at all.*” Ricky (P3) continued to describe how he often unknowingly went over his expected commitment of 1 hour per session: “*I know*[the ERO] *starts her day at 12:00* [p.m.]. *So, 12:00 till 2:00, but you come in, you don’t just do an hour here. I’m sometimes here for an hour and a half. I’ve looked, ‘Oh, shit, I’ve got to go.*” Carl (P4) described how he would rearrange his schedule, where possible, in order to be able to attend peer sessions: “*When I can rearrange things, and that, I say… this is like top of my list, because I enjoy it. Something else can be moved. I’ll put this first.*”

EROs and clients also cited the central importance of peers being able to act as an “*extra pair of eyes.*” As highlighted below by Sigourney (ERO1) and Agnes (C5), respectively:
It just makes life so much easier that, if I’m with somebody who’s… either on an induction or they’re brand-new to the gym and I’ve got to spend a bit more time helping them, I can just ask whatever volunteer’s in on the day to keep an eye on a certain person if they’re a bit wobbly.
*I think, because if you can’t find* [ERO], *the volunteers are usually there, and you just have to shout, you know, and they’ll come across and help you with whatever you need to do.*

Notwithstanding, acceptability and peer engagement differed in some cases related to the demographic characteristics of the peers. Doris (C1) expressed preference for a younger and/or female peer: “*I’m more in tune with the younger ones. Erm… and possibly female as well.*” Tegan (C3) also reported a preference for a female peer due to a recent negative personal experience:
*Well, me, in particular, would feel better speaking to a female. Especially after what’s happened to me a couple of weeks ago. So, that’s where I am at the moment. But there’s nothing wrong with male volunteers. That’s what I feel comfortable*[with] *at the moment because of what I’ve been through the last couple of weeks.*

Jessica (C4) discussed her positive experience interacting with a peer of a similar age: “*I’d say he probably is, ish, my age group. But I wouldn’t like to comment* (laughs). *So, that’s probably why I can relate to him better because he’s not old and he’s not young. He is kind of my age.*” Sigourney (ERO1) recounted an instance where a peer and client bonded over a shared medical condition: *“They got talking and they found out that they had similar conditions. So, that was actually quite good because* [Paul]*was actually explaining to this gentleman about how he… what he’d done and how it had helped him.*”

Similarly, Ricky (P3) detailed how he had personally benefitted from discussing details of his medical background with a client on the scheme:
*I said, “I’m a diabetic, I’m a type II diabetes, and I’ve suffered with multiple blood clots.” And then, they’ll say to me, “Oh, do you take warfarin?” “Yeah, yeah, I take warfarin.” One lady has actually said to me, “Go to your doctor.” She did say to me, she did write it down, I’ve put it somewhere in* [there]*… “Go and ask about…” a completely different drug. Now, that’s come off one of the ladies here.*

### 3.3. Practical Considerations and Lessons for the Future

Participants highlighted modest and variable client attendance of peer sessions. Carl’s (P4) peer sessions had the best attendance: “*I’d say it probably averages… at any one time, I’d say between seven and nine, as a guesstimate,*” though attendance varied around three to six clients among other peer sessions. Ricky (P3) attributed a drop in attendance to seasonal variation:
*I did mention this once to* [ERO]. *I said, “Wouldn’t it have been…” I’ve got to be careful how I said this on your Dictaphone… “Would it be more in your interest to… probably introducing this in spring, so you’d be going through the summer months?” Because when I used to come in, in summer months, it was busy.*

Moreover, Sigourney (ERO1) believed the drop-in nature of supervised sessions contributed to the variable and unpredictable attendance, highlighting the difficulty of trying to ensure peers were implemented within the busiest sessions: “*Monday morning was manic, and I didn’t have any volunteers in because the volunteer usually comes in on a Monday afternoon, but the afternoon session was a lot quieter because a lot of them had come in, in the morning.*” Nonetheless, peer attendance remained generally consistent. Peers cited a deterioration of health related to their medical condition(s) as the primary contributing factor for any nonattendance. As explained by Paul (P2):
The only time I don’t come in is when my Parkinson’s is really bad. And I know I’m not going to be able to speak particularly well because the brain and the mouth don’t work sometimes. Erm… which can be a bit embarrassing for me, and I don’t want to put my embarrassment onto other people.

Sadly, one peer stepped back from their role for an extended period of time due to the unexpected loss of a member of their immediate family. Owing to the sensitive and unpredictable nature in respect of if, or when, the peer may wish to return, EROs were asked to try to identify another prospective peer to cover these sessions, though no suitable replacement could be found. Unfortunately, this meant despite a positive early experience, Tegan (C3) did not receive continued peer support throughout her involvement with the ERS: “*I haven’t seen anyone for ages. Because I thought they’d all been like told not to come or something, because I haven’t seen anyone. The only person I’ve seen is* [the ERO].” 

Participants also highlighted a lack of clarity in relation to the specific roles and responsibilities of peers, as well as their professional backgrounds and expertise. Agnes (C5) acknowledged that peers had completed ERS themselves prior to assuming the peer role and expressed how this brought her comfort: “*Obviously, she* [the peer] *knows… if I’m just starting out, she kind of knows what, you know, I’m going through. So, it’s just kind of making you feel comfortable.*” However, Tegan (C3) was unsure whether peers had sufficient experience and knowledge to positively enhance her experience:
*I don’t know what training, if they’ve been trained. I don’t know. I was just introduced to a volunteer. And then she* [the peer] *said that she used to be on the scheme. So, I thought to myself that she only knows what I’m going to know after 12 weeks.*

Doris (C1) expressed similar uncertainty regarding peers’ backgrounds when asked whether peers were suitable for their roles:
Yes. Because I understand that they’ve been there in front of us. I think they’re doing what they’ve experienced, like we are. And, they’ve had promotion to do the job they’re doing. I think. That’s right, isn’t it? We might know more than them.

Mabel (P1) expressed a lack of confidence when describing an interaction where a client had asked about the nature of her role as a peer: “*I do say, I’m just here to talk to people. To make them feel comfortable and for them to want to come back and use the gym. That is, isn’t it? There isn’t anything else I should be doing?*” Peer responses belied a lack of clarity surrounding the parameters of the peer role when asked to discuss the adequacy of the training they had received. Mabel (P1) believed the training to be sufficient, though inferred that she would feel more confident with enhanced knowledge of how some of the exercise machines worked: “*I don’t know how all the machines work… but then again, I don’t think that’s my role, is it? Mine is just to support people and make them happy and want to come in, and I think that you did that.*” Positively, Ricky’s (P3) interpretation of the peer role was more closely aligned with how it was intended:
*The role you’re asking of me is not… it’s not a severely demanding role. It’s not something where… there’s no more amount… you know, what other amount of training can you give me?… Can you give me the fact that I might end up working in the same capacity as*[the ERO]*? I don’t think that would be any beneficial to me. It wouldn’t be any beneficial to you.*

Moving forward, Sigourney (ERO1) suggested that group-based peer training would collectively improve the standardisation and comprehension of peer roles: “*Yeah, more structure to it. Rather it just sort of be left as an individual thing. I think it could do with more of a group* [training]*. I think that’s how we would end up with a better volunteer scheme.*” 

## 4. Discussion

Overall, ERS clients, EROs, and peers indicated good acceptability of the social-identity-informed peer support ERS intervention. EROs and clients valued peers as a means to reduce the burden on EROs and as readily available sources of proxy social support. Peers themselves discussed how the role provided a personal sense of fulfilment and satisfaction, though client perceptions of acceptability varied according to perceived similarity of peer gender and age. Primarily, peers consistently attended scheduled peer sessions; however, peer sessions had modest and variable client attendance. EROs and peers attributed this to seasonal variation, recommending future peer support interventions to be scheduled earlier in the year during spring or summer. A lack of clarity existed among peers and clients regarding the nature and extent of peer knowledge and expertise. 

EROs utilized peers appropriate to the perceived needs of clients in the gym during supervised sessions. Severity of health condition(s) and stage of scheme influenced perception of need. Peers primarily offered support via nontechnical guidance on how to operate exercise equipment and social support. For instance, peers provided experiential advice on how to use exercise equipment in ways designed to stimulate engagement and manipulate the experience of time. The ability to disseminate experiential knowledge is a fundamental appeal of peer support interventions and repeatedly cited as a facilitator of participant engagement [28,29]. In rare instances, peers offered technical guidance on the use of exercise equipment, such as which machines to use or how to use certain equipment. However, EROs quickly identified and rectified such undesirable behaviour. Clearly defined peer role boundaries are critical to the success of peer support interventions [30,31] and frequently cited as vitally important among professionals tasked with supervising peers [32]. Positively, the ERO described this as an isolated incident that received swift resolution. In all cases, peer social support primarily consisted of extending warm welcomes upon client entry to the gym and circling around the gym environment intermittently during sessions to check that clients were content. Alongside direct peer support, clients also discussed how “just having peers there” provided feelings of reassurance. In line with Bowe et al. [33], the presence of peers facilitated enhanced perceptions of social support and safety. Feeling adequately supported and safe are prerequisites for clients to feel confident enough to attend future exercise sessions independently and reach out to other social groups [34]. 

Evidenced by ERO and peer interviews, peers varied in relation to their interpersonal skills and confidence approaching clients. These characteristics fall within a broader range of peer professional backgrounds, experiences, personality, and illness characteristics that have been linked to peer support successes [35]. Lack of confidence, or shyness, of peers can be detrimental to peer implementation success, relative to peers who are considered to be warm, energetic, and humorous [36]. Lorthios-Guilledroit et al. [36] found that peers with extensive prior experience in facilitating group discussions were more confident and self-efficacious in their ability to utilise these skills during peer-initiated group discussions. This finding is in parallel to the current study, where peers with professional backgrounds associated with high-level interpersonal skills (e.g., taxi driving) were more confident in their ability to approach and engage new clients. As in the study of Holman et al. [35], ERS peers appeared to have skills in different areas, which influenced their approach to the role. Intriguingly, EROs highlighted subtle differences between peers, emphasising distinct individual strengths of peers. Thus, the findings of this study suggest that there may be advantages of employing peers with varying approaches and communication styles to complement the demographically heterogenous ERS context.

Clients were acceptable to the peer role, citing the presence of peers as making them feel more comfortable. The recruitment and implementation of ERS peers who were managing health conditions provided a platform for relatability, enabling peers to serve as positive role models [37]. Peers’ own experiences of health conditions distinguished them from EROs, promoting recognition of peers as “normal people” whom clients were able to talk to on an equitable footing [36,38]. Accordingly, client discussions with EROs were typically briefer and focussed on receiving technical or health-related guidance. In contrast, interactions with peers were focussed on client general well-being and covered a variety of non-expert topics, such as the weather, current political events, and television. One peer described this as “bringing the living room into the gym,” an approach that is likely to have facilitated the positive perceptions of comfort referenced by clients. The ability of ERS peers to contribute to enhanced perceptions of comfort represents a meaningful, positive consequence of their implementation. Comfort is positively associated with social identification and a sense of group belonging [39], presenting preliminary evidence of ERS peers’ ability to facilitate shared social identity. Moreover, feelings of comfort positively predict participant adherence among peer support interventions [38]. Subsequently, wider implementation of ERS peers may mitigate the pervasive and long-standing issue of high ERS dropout [1,38]. Notwithstanding, some clients did express preference for peers of the same age and/or gender as themselves. Age and/or gender are observable characteristics upon which perceptions of similarity can be based, thus serving as rudimentary criteria for social identification [19]. As such, age and/or gender positively influence peer–recipient relationships [37], though the importance of shared peer demographic characteristics varied among clients [35]. ERS clients and peers also recounted positive experiences stemming from shared health conditions and treatments. Shared demographic characteristics further enhance perceptions of comfort and collective understanding, thus promoting peer acceptability [37]. Consequently, where possible, it is advisable for ERSs to recruit peers who vary in regard to age, gender (within the range of 45–70 years), and health condition. By doing so, EROs will be able to assign peers to clients who are demographically comparable, whilst simultaneously providing clients with autonomy over which peers they subjectively perceive to be most similar to themselves.

Clients and EROs cited the benefits of peers supporting existing ERO roles. Peers brought fresh insight and enthusiasm to the ERS, contributing to a fun and engaging exercise environment [40]. Moreover, the presence of peers enhanced the volume and perceived accessibility of social support, surpassing that which could be offered by EROs alone. Accordingly, the integration of ERS peers can minimise the likelihood of client dissatisfaction with inadequate social support opportunities. This is a promising finding given the greater ERS dropout risk associated with inadequate social support [12,15,16]. Consistent with previous peer support health initiatives [41], ERS peers presented an affordable means to enhance perceptions of support relative to the implementation of additional trained EROs. Positively, peers were satisfied and fulfilled by their roles. In line with the study of MacKean et al. [37], ERS peers valued their role as an opportunity to “give something back” by sharing experiential knowledge with clients. Further, whilst sharing of such knowledge has the evident potential to be beneficial to clients, it also represents additional progression among peers on their journey from someone that is currently managing an existing health condition. Through sharing experiential knowledge, peers are able to reinterpret a negative life experience (i.e., the emergence of a health condition) into a positive life experience by sharing details of their story which may ultimately benefit others in similar situations [42]. Similar to the study of Stevens et al. [41], the opportunity for peers to witness client progress and to be seemingly appreciated for their support appeared to be a fundamental factor for long-term peer engagement. Moreover, positive peer accounts indicate acceptable pitching of the peer role in regard to expected short- and long-term peer commitment and level of peer responsibility. 

Participants described modest and variable client attendance of peer-supervised sessions. Occasionally, the “drop-in” nature of sessions prohibited the ability to schedule peers to attend the busiest supervised sessions. Subsequently, whilst peers represent a cost-effective source of additional social support, the efficiency and cost-effectiveness of their implementation may be enhanced via more restrictive scheduling of available ERS sessions—that is, reducing the availability of “drop-in” sessions to ensure clients are attending sessions in which peers are present. Peers described instances where they attended sessions with few clients. It is likely that repeated occurrence of poorly attended sessions may be detrimental to ongoing peer engagement and motivation. Nonetheless, peer attendance remained generally high throughout, with absences primarily explained by temporary deterioration of peer health. Unsurprisingly, this a consistent finding among the peer support literature within the health domain [36,43,44]. This study reinforces the advice of Holman et al. [35] to recruit a sufficient number of peers to cover short-term absences. ERS providers should weigh the increased economic costs of recruiting additional peers with the extent to which peers contribute to positive ERS client experiences and adherence behaviour. However, both perceived and actual costs and benefits of ERS peer implementation are likely to vary as a consequence of inter-ERS heterogeneity. Still, ERSs can benefit by implementing the peer support strategies described within this study. Moreover, the flexibility of the peer approach enables tailoring to accommodate differences in individual ERS characteristics and available financial resources. For instance, ERSs with sufficient financial resources can mitigate the negative implications of unpredictable peer absences by recruiting a greater number of peers than was possible here. In contrast, ERSs with limited financial resources may be unable to afford to incentivise peers in a similar capacity as the current study, and thus, it may be unrealistic to expect peers to perform the same quantity of roles. In such cases, peer roles, and subsequent means of incentivisation, may be reduced proportionately to facilitate the recruitment of a sufficient number of peers to cover unexpected absences. Ultimately, this study demonstrates that even basic roles performed by peers can positively enhance ERS clients’ experience by promoting increased feelings of comfort and belonging. Therefore, it is recommended for ERSs to use the guidance provided here to explore the viability of embedding structured peer support within their existing provision.

Clients and peers also highlighted the need for greater clarification of the peer role. Clear delineation of peer roles enables peers to have confidence in the roles that are expected of them [42] and enables clients to fully utilize all aspects of peer support. In line with the study of Gillard et al. [40], whilst the training protocols utilised within the current study were pragmatic, it is apparent that they failed to sufficiently facilitate shared expectations of the peer role. The consequences of such were discussed earlier, where some peers attempted to offer technical guidance on exercise machine use. Moving forward, greater consensus of peer roles may be achieved via group-based peer training. Group-based peer training can promote greater standardisation and allow successful strategies to be shared among peers [35]. Alas, there also appears a need to more clearly outline the nature and scope of peer roles among ERS clients. The ERS peer support intervention sought to promote equitable peer–client relationships to facilitate perceptions of sameness and promote the adoption of social identity. The conceptualisation of peers as sharing “sameness” with the intended recipients of peer support is critical to their effectiveness [45], and attempts to elevate peer status above clients via enhanced technical knowledge or expertise is likely to have been met with resistance [41]. However, some ERS clients appeared to undervalue the roles of peers if they interpreted peers as “only knowing what I’ll know when I finish the scheme.” Thus, providing greater description of peer roles and the intended value of peers among new ERS clients may further enhance their acceptability. 

### Limitations

The use of EROs to recruit interview participants may be considered a limitation of this study. It is possible that EROs selectively sought to recruit clients known to have had positive ERS peer experiences and proclivity for sharing details of these experiences with others. Notwithstanding, EROs were purposefully provided with only brief and vague details of the content and focus of the client interviews to mitigate selection bias. The proposed minimal influence of such bias is supported by clients’ willingness to share their views regarding how the intervention could be enhanced and to describe rare instances of dissatisfaction. In addition, the presented demographic data for clients and peers were collected as part of a separate quantitative evaluation of the ERS. The quantitative evaluation utilised existing ERS data collection procedures that did not include participant educational level. Such data may have complemented this study’s findings, particularly in relation to peers’ knowledge and comprehension of their intended roles as described during peer training. The study may also have benefitted from recruiting more ERS client perspectives. The scheme experienced poorer adherence of peer sessions in the months immediately preceding and following the Christmas 2019 period, limiting opportunities for data collection. Further, the scheduled data collection period came to an enforced end in March 2020 due to the emergence of the COVID-19 pandemic. Nonetheless, the study provides acceptability findings from a variety of key ERS stakeholders, including a diverse range of client perspectives across two delivery sites. These findings provide guidance on the development and implementation of structured peer support interventions for ERSs. Thus, it is unclear to what extent additional client interviews may have yielded additional salient knowledge.

## 5. Conclusions

This study presents retrospective acceptability findings from a 12-week ERS social-identity-informed peer support intervention. EROs, clients, and peers reported good acceptability on the integration of peers within the existing scheme structure. Subsequently, this study demonstrates that contextually appropriate, acceptable social-identity-informed peer-based interventions can be developed and implemented within ERSs. Further, this study operationalises existing knowledge regarding the pre-eminent value of social support among ERS participants and describes the practical application of a social identity approach to ERS provision using peers. The structured basis of the peer support intervention displaces the onus from individual ERS clients having to seek and instigate social interaction, improving the equitability of perceived social support opportunities, particularly among those with social anxiety and/or inadequate interpersonal skills. Moreover, the flexible nature of peer support interventions enables this study’s findings to be transferred across the heterogenous ERS landscape. Accordingly, whilst specific roles and recruitment practices may require modification for use in different ERS contexts, this study provides useful guidance in relation to valued peer attributes and roles, and ways to maximise acceptability among various ERS stakeholders. Future exploration should seek to identify variable ERS effectiveness among schemes that incorporate an embedded social support component and those that do not. In addition, subsequent quantitative evaluation can supplement this study’s findings by furthering understanding of acceptable ERS peer roles among a larger cohort of ERS clients. Such investigation will enable continued refinement of peer roles appropriate to client needs.

## Figures and Tables

**Table 1 ijerph-18-01720-t001:** Key demographic characteristics of peers and clients.

Pseudonym (ID Number)	Age (Years)	Gender	Medical Condition	BMI (kg/m^2^)	Peer Sessions Attended (%)
Peers					
Mabel (P1)	67	Female	Respiratory	28.7	73% (11/15)
Paul (P2)	67	Male	Parkinson’s	29.4	93% (14/15)
Ricky (P3)	59	Male	Diabetes	36.6	69% (11/16)
Carl * (P4)	44	Male	Mental Health	31.4	100% (5/5)
Meredith^‡^	57	Female	Gastrointestinal	19.8	47% (8/17)
Clients					
Doris (C1)	65	Female	Musculoskeletal	32.6	N/A ˆ
Harry (C2)	77	Male	Cancer	26.4	N/A ˆ
Tegan (C3)	49	Female	Mental Health	31.2	N/A ˆ
Jessica (C4)	56	Female	Cardiac	21.8	N/A ˆ
Agnes (C5)	75	Female	Cardiac	27.1	N/A ˆ

* Recruited in November 2019, ^‡^ not interviewed, ˆ data not recorded, BMI = body mass index.

## Data Availability

The data presented in this study are available upon reasonable request from the corresponding author. The data are not publicly available due to ethical restrictions.

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
