# Peer review of "Multi-Stakeholder Retrospective Acceptability of a Peer Support Intervention for Exercise Referral"

_ijerph, 2021, doi:10.3390/ijerph18041720_

Round 1
Reviewer 1 Report
The authors answered adequately to the previous comments.
Author Response
No further comments to address. Thank you again for your constructive comments on the previous version of the manuscript.
Reviewer 2 Report
Dear Authors,
You have done good work to improve the manuscript.
But your manuscript still need correction of the "Bibliography" chapter. I have marked all items in the attached pdf version.
Reviewer 2

Author Response
Thank you for your comments on the previous draft of the manuscript. The required changes to the bibliography have been made. All relevant journal names are now abbreviated.
Reviewer 3 Report
This manuscript is a report on a qualitative investigation of multiple stakeholders’ perceptions of a peer support intervention for exercise referral. The topic falls squarely within the domain covered by this journal. Examination of the acceptability of the intervention from the perspective of the people directly involved in the delivery and receipt of the intervention is a desirable aspect of the study. The manuscript is very well-written. These positive impressions notwithstanding, I have several concerns about the manuscript in its current form:
- On l. 39, it should be “proposed.”
- On l. 55, it should be “championed.”
- Regarding the material on l. 110-114, what is the difference between “non-specialist assistance using exercise equipment” and “specialist advice on how to use or operate gym equipment”?
- On l. 202, it should be “Agnes (C5).”
- The issue highlighted on l. 368-375 represents a potential limitation of the peer support approach, although the provision of such support could be further incentivized if deemed of sufficient value by those controlling the incentives.
- Perhaps the most important finding of the study—the need for role clarity among peer volunteers—is emphasized appropriately in the Discussion section. It seems that the optimal role and training of peer volunteers is really an empirical question, one that should be examined experimentally in future research with larger samples. It might be helpful to incorporate cost-effectiveness into such investigations so that the extent to which investments in peer volunteer training and resources produce outcomes that offset health care costs.
Author Response
Please see the attachment

This manuscript is a resubmission of an earlier submission. The following is a list of the peer review reports and author responses from that submission.
Round 1
Reviewer 1 Report
The study proposes social support as an important premise to foster participation and satisfaction in ERS. Overall, the design of the study matches the purposes of the study, but there are some issues that must be addressed.
The authors refer that one of the ims of the study was to identify the factors contributing for social support. But in the results and the discussion a clear list of explanatory factors does not emerge.
The theoretical frame work used to build the interviews was not presented in the introduction, nor in the methods section. How was the script elaborated? what questions were made? it was the same script for supervisors, volunteers and users?
The procedures are not presented. When were the interviews made? After the 12-weeks period? During it?
The questions above bring the question of the interpretaion of the data. Was it deductive or inductive?
The authors aknowledge that the composition of the sample can be a limitation. The volunteers were a convenience one, and that represents a potential bias.
Regarding the users, 4 were female and one male. Probably it had some influence on the results. The educational level of the participnats was not disclosed and I beleive it can be helpful for the discussion.
Reviewer 2 Report
Manuscript ID: ijerph-1030377
Type of manuscript: Article
Review of the manuscript entitled “Multi-stakeholder retrospective acceptability of a peer support intervention for exercise referral“:
There is no clear justification for the purpose of this study in the introduction.
The consent of the bioethics committee is described very laconic (the lack of precise data and the lack of the consent of the participants).
The results are illegible. They are hard to read. Perhaps you will consider some tips described in literature (Silverman, D. 2006: Interpreting qualitative data: methods for analyzing talk, text and interaction, third edition. London, Thousand Oaks: Sage). You can mark a cited sentences, e.g. in italics, or make tables (after all, they are to be evidence in the questions / topics posed in the line 71-74).
The discussion is an extensive literature description, so consider limiting it to the obtained results.
Please, re-read and consider proofreading for the reader's.
I noted also some editorial errors in the manuscript pdf.
